# First Comprehensive Study of a Giant among the Insects, *Titanus giganteus*: Basic Facts from Its Biochemistry, Physiology, and Anatomy

**DOI:** 10.3390/insects11020120

**Published:** 2020-02-12

**Authors:** Jiří Dvořáček, Hana Sehadová, František Weyda, Aleš Tomčala, Markéta Hejníková, Dalibor Kodrík

**Affiliations:** 1Institute of Entomology, Biology Centre, CAS, Branišovská 31, 370 05 České Budějovice, Czech Republic; reditel@cervenydvur.cz (J.D.); sehadova@yahoo.com (H.S.); marketahejnikova@seznam.cz (M.H.); 2Faculty of Science, Branišovská 31, University of South Bohemia, 370 05 České Budějovice, Czech Republic; weydafk@seznam.cz; 3Faculty of Fisheries and Protection of Water, CENAKVA, University of South Bohemia, Institute of Aquaculture and Protection of Waters, Husova tř. 458/102, 370 05 České Budějovice, Czech Republic; a.tomcala@centrum.cz

**Keywords:** brain, Cerambycidae, chromatography, compound eye, digestive enzyme, electron microscopy, lipid metabolism, muscle, sensilla, testes

## Abstract

*Titanus giganteus* is one of the largest insects in the world, but unfortunately, there is a lack of basic information about its biology. Previous papers have mostly described *Titanus* morphology or taxonomy, but studies concerning its anatomy and physiology are largely absent. Thus, we employed microscopic, physiological, and analytical methods to partially fill this gap. Our study focused on a detailed analysis of the antennal sensilla, where coeloconic sensilla, grouped into irregularly oval fields, and sensilla trichoidea were found. Further, the inspection of the internal organs showed apparent degeneration of the gut and almost total absence of fat body. The gut was already empty; however, certain activity of digestive enzymes was recorded. The brain was relatively small, and the ventral nerve cord consisted of three ganglia in the thorax and four ganglia in the abdomen. Each testis was composed of approximately 30 testicular follicles filled with a clearly visible sperm. Chromatographic analysis of lipids in the flight muscles showed the prevalence of storage lipids that contained 13 fatty acids, and oleic acid represented 60% of them. Some of our findings indicate that adult *Titanus* rely on previously accumulated reserves rather than feeding from the time of eclosion.

## 1. Introduction

*Titanus giganteus* (Coleoptera: Cerambycidae: Prioninae) is supposedly one of the largest living representatives of the class Insecta. It was described by Linnaeus in 1771, and for many years was considered a great rarity, rarely represented in collections [1,2,3]. Increasingly higher numbers of very large specimens of this species have begun to appear in the last 30 years [4]; yet, even today their value among collectors is great, and the combination of unavailability and extraordinary pricing has been one of the main reasons why this species has not been studied in detail so far.

The distribution area of *T. giganteus* is currently listed as Colombia, Guyana, French Guiana, Suriname, Peru, Ecuador, Bolivia, and the northern part of Brazil [5,6], indicating this beetle is restricted to the Amazon rainforest. Although an increasing number of individuals are found each year, the research status of this species remains unchanged. Its mode of life, host plant (the only mention is in [6]: *Siparuna pachyantha* A.C. Sm. as a host plant), and developmental period are thus unknown. Interestingly, the *Titanus* larva is not known at all.

All historical scientific reports about the description of this species to date have solely focused on morphology [1,7,8,9,10], with only a few references to bionomy [11]. Anatomical and physiological investigations are missing, although there are some simple basic questions: Can a model of the *Titanus* life history possibly drawn from knowledge of their basic anatomy and physiological parameters? Is there anything different in the body of a huge insect when compared with ordinarily sized insect bodies?

Even for other groups of longhorn beetles, representatives of the Cerambycinae and Lamiinae subfamilies, information on insect physiology and internal anatomy is limited. Most of these studies deal with the digestive system function in larvae of the pest species; works on adults are rare. Thus, there are studies that describe the gut microflora and its importance in digestion [12,13]; the activity of amylases [14,15], carbohydrases [16], and proteases [17,18,19]; and methods of cellulose digestion [20,21,22,23]. The digestive tract is also the most described part of the internal anatomy of longhorn beetles [24,25].

The comparison of the above-mentioned data with those available on the Prioninae subfamily is complicated. Generally, Cerambycinae and Lamiinae larvae utilize a varied plant diet with different nutritional composition, and in many species, adults also ingest food, at least in the form of maturation feeding [26]. However, it is typical for Prioninae that adults do not accept food (apparently except water)—not even as maturation feeding. Larvae feed on dead wood often infested by fungi; the adults exhibit night activity (i.e., a sensory system adapted thereto), and at least the males of most species have the capability of flight (therefore, they must have access to an adequate source and supply of energy).

Unfortunately, there is a lack of information for the whole Prioninae group in general. Most of the published studies are taxonomic in nature or refer to geographic distribution, and very few articles have been written about the life or physiology of Prioninae species; for instance, there are articles on xylomycetophage *Eoxenus relictus* [27,28], the gastrointestinal tract and reproductive system of the New Zealand *Prionoplus reticulatus* [29,30], the gastrointestinal tract of the African *Macrotoma palmata* [31], and the North American *Prionus laticollis* [32,33], identification of xylanase from symbiotic bacteria in *Prionus insularis* larvae [21], and identification and characterization of a pheromone from *Prionus californicus* adult females [34].

A closer look at *T. giganteus* can provide more information about this very old subfamily, particularly since the body size of *Titanus* is close to the physiological limits imposed by today‘s atmospheric composition [35]. The length of the largest reliably documented individual is 167 mm [4]. Two other giant species of beetle *Dynastes hercules* (L., 1758, Dynastinae) and *Macrodontia cervicornis* (L., 1758, Prioninae) attain a similar size, including extremely long mandibles or narrow protuberance on the head or shield. However, *Titanus’* size is made up almost exclusively of its large body. 

We had a unique opportunity to analyze one big individual of *T. giganteus,* and thus revised more or less known morphological and anatomical data, but mainly focused on the beetle’s ultrastructure of its sensory organs, and on analyses of biochemical features of the gut and the muscle system to characterize possible energy sources for flying. For that we processed the entire beetle body and analyzed it by relevant methods available in our laboratory to obtain as much data as possible. We are aware that one individual is not sufficient for a deep ground-breaking study, on the other hand “destructive” biochemical, physiological, and anatomical analyses of larger set of this rare beetle are not possible. With regard to this reality we believe that despite this limitation our study brings interesting data and extends our knowledge about this extraordinary creature.

## 2. Materials and Methods

### 2.1. Titanus Origin

The analyses were performed on a male *Titanus giganteus* collected 12 January, 2019 in French Guiana in Montagne de Kaw, Camp Patawa. The male was attracted to light (250 W) at 1.30 h after midnight. The male (weight 37.5 g, length 155 mm) was assayed within 10 days from the time of capture. 

### 2.2. Dissection

The animal was decapitated using of surgical scissors under the deep carbon dioxide anesthesia, all legs and wings were removed and the head and thorax with attached abdomen were investigated separately. All internal organs were dissected under the phosphate buffered saline (PBS) and immediately photograph by digital cameras Nikon D7200 equipped by objective Tamron SP 90 mm, F/2.8, Di Macro 1:1 and Sony equipped by Sony macrolens SEL 30 mm, F/3,5 macro 1:1.

### 2.3. Mallory Staining and Microscopy of Testes

Dissected testicular follicles were fixed overnight at 4 °C in Bouin–Hollande solution without acetic acid but supplemented with mercuric chloride [36]. Standard techniques were used for tissue dehydration, embedding in paraplast, sectioning to 7 μm, deparaffinization, and rehydration. The sections were treated with Lugol’s iodine followed by 7.5% solution of sodium thiosulphate to remove residual heavy metal ions, and then washed in distilled water. Staining was performed with HT15 Trichrome Stain (Masson) Kit (Sigma) according the manufacture protocol. Stained sections were dehydrated, mounted in DPX mounting medium (Fluka), and viewed and imaged under microscope Olympus BX51 equipped with CCD camera (Olympus DP80).

### 2.4. Scanning Electron Microscopy—Sample Preparation

For electron microscopy analyses the *T. giganteus* external organs were cut-off by a scissors or scalpels, dehydrated in a series of ethanol solutions, dried by critical point drying techniques (CPD), glued on aluminum holder, sputter-coated with gold and observed and photographed using a HR SEM Jeol 7401F scanning electron microscope. 

### 2.5. Digestive Enzyme Activity Determinations

The activities of amylases, proteases and lipases were determined in the midgut and hindgut together. The organ was homogenized (sonicated) in appropriate buffer (see below), centrifuged and the aliquot 0.05 equiv. tested for enzyme activity.

*Amylase assay.* The assay was performed with 3,5-dinitrosalicylic acid reagent (DNS) according to [37] as modified by [38]. Briefly, 25 μL aliquot sample extract in 100 mM phosphate buffer [39] pH 5.7 was mixed with an equal volume of 2% soluble starch made of the same buffer with 40 mM NaCl. The reaction mixture was incubated at 30 °C and under constant agitation for 40 min until it was terminated by adding of 200 μL DNS. Then the solution was heated at 100 °C for 5 min, cooled, clarified by centrifugation (10 000 g 10 min), and the absorbance was read in supernatant at 550 nm. Enzyme activity was calculated in μmol maltose per mg of fresh organ weight.

*Protease assay.* The protease activity was assessed with the resorufin-casein kit (Roche) according to manufacturer’s instructions. Briefly, 20 μL sample extracts in 0.2 M tris pH 7.8 were mixed with 20 μL of 0.4% substrate (resorufin-casein) and 20 μL of 0.02 M calcium chloride solution, and adjusted up to 100 μL by the same tris buffer in the microtubes. The mixture was subsequently subjected to gentle shaking for 1 h at 37 °C. The reaction was terminated by addition of 240 μL of 5% trichloroacetic acid, and after 10 min of subsequent incubation at 37 °C centrifuged to remove the non-hydrolyzed casein. The absorbance was measured at 490 nm. Appropriate controls without the samples were assayed simultaneously. Protease activity was expressed in units of proteolytic activity per mg of fresh organ weight; this unit (U) was defined as the amount of enzyme (mg) which caused an increase in optical density by 0.1 per min in 1 mL of the reaction mixture [40].

*Lipase assay.* The lipase activity was assessed with 4-methylumbelliferyl butyrate (4-MU butyrate) according to [41] as modified by [38]. Five microliters of 2 mM substrate diluted in dimethylsulfoxide (DMSO) were added to microplate wells with organ extracts in 0.2 M tris pH 7.8 diluted to 195 μL with 100 mM phosphate buffer pH 5.0 [39]. Samples were incubated at 30 °C and the release of the fluorescent 4-methylumbelliferone (4-MU) was monitored at 5 min intervals at 327 nm excitation and 449 nm emission with a Synergy 4 multi-mode microplate reader (BioTek Instr., Winooski, Vermont). Activity was expressed in pmol of 4-MU/min/mg of fresh organ weight.

### 2.6. Mass Spectrometry Determination of Muscle Lipids

High performance liquid chromatography coupled to electrospray ionization tandem mass spectrometry (HPLC-ESI MS/MS). Lipids were extracted from *T. giganteus* flying muscles with chloroform and methanol solution (ratio—2:1) following the method of [42]. Samples were homogenized in extraction solution with glass beads using TissueLyser LT mill (Qiagen, Hilden, Germany). Homogenates were dried and resolved in 500 µL of chloroform and methanol (1:2) with internal standard phosphatidyl choline (PC) C17:0/C17:0 (Sigma Aldrich). The sample aliquots (5 µL) were injected by the autosampler Accela (Thermo Fisher Scientific, San Jose, CA, USA) and separated on the Gemini column 250 × 2 mm; i.d. 3 µm (Phenomenex, Torrance, CA, USA). The mobile phase consisted of (A) 5 mmol/L ammonium acetate in methanol, (B) water, and (C) 2-propanol. The analysis was completed within 80 min with a flow rate of 200 µL/min by following gradient of 92% A and 8% B in 0–5 min, then 100% A till 12th minute, subsequent increasing the phase C to 60% till 50 min and hold for 15 min and then in 65th minute back to the 92:8% A:B and 10 min to column conditioning. The column temperature was maintained at 30 °C. A linear ion trap LTQ-XL mass spectrometer (Thermo Fisher Scientific) was used in both positive and negative ion ESI mode. The settings of the system followed the methodology published earlier [43]. Particular lipid species were determined based on m/z value, retention time, behavior in positive and negative ionization mode, and characteristic fragmentation pattern. The data were acquired and processed using Xcalibur software version 2.1 (Thermo Fisher Scientific, San Jose, CA, USA) (for details see [43]). 

*Gas chromatography*—*flame ionization detection (GC-FID)*. The lipid extracts for fatty acid (FA) analyses were methylated according to the methods of [44]. FA composition was analyzed by gas chromatography (GC) (Trace Ultra FID; Thermo Scientific, Milan, Italy) using a BPX-70 50 m fused silica capillary column (id. 0.22 mm, 0.25 μm film thickness, SGE, Austin, Texas, USA). The temperature gradient starts at 70 °C and holds for 30 s, then the temperature rises by 30 °C per minute till 150 °C. After that, the temperature rises to 220 °C by a rate of 1.5 °C per minute and holds for 11 min. The whole analysis takes 60 min. The peaks were identified by comparing sample retention times to retention times of the standard mixture Supleco 37 Component FAME mix (Sigma-Aldrich). The data were acquired and processed using Xcalibur software as mentioned above. 

### 2.7. Data Presentation

The photo plates were done with Adobe Photoshop software. The graph results were plotted using the graphic software Prism (Graph Pad Software, version 6.0, San Diego, CA, USA). 

## 3. Results

### 3.1. Brief Summary of External Morphology

The body structure of *Titanus* shares common features with that of other members of Prioninae (Appendix A). The studied *Titanus* was 155 mm long with a weight of 37.5 g. The head possessed thick long antennae reaching half the body length, very robust mandibles, and thinner maxillary and labial palps. The large compound eyes (see also below) occupied a large proportion of the head that was 35 mm in length (including mandibles) and 23 mm in width. The wingspan was 250 mm and exceeded the elytron span by approximately 20 mm. The abdomen was very flat, especially in the marginal parts where spiracles were visible. From a total of eight spiracle pairs, six were large; the seventh was smaller but functional, and the eighth was only rudimentary.

### 3.2. Analyses of Selected Sensory Organs

*Compound eyes*—The compound eye of *T. giganteus* contained hundreds of mostly hexagonal ommatidia (Figure 1A). Ommatidia located in the central region of the eye were covered by typical hexagonal facets (Figure 1B), while facets on the periphery of the eye were often irregular, pentagonal, and even square (Figure 1C). No ommatrichium (interfacetal hairs) was found on the smooth surface of the eye. Scanning electron microscopy of the cross section of the eye partially revealed the inner structure of the ommatidium. Each ommatidium contained a corneal biconcave lens and an undetermined number of retinula and pigment cells (Figure 1D,E). It looks that a clear zone separating the dioptric apparatus from the photoreceptive structures was not developed, and the eye thus resembled an apposition eye. Axons of retinula cells of each ommatidium exited the eye as distinct bundles through holes in the basement membrane (Figure 1E,F), and are linked with their target neurons in the lamina. Nerve bundles are surrounded by supporting cells (Figure 1F) that create a sleeve, and are intertwined with a tangle of muscles and tracheas (Figure 1G,H).

*Antennal sensilla*—Antennae form an important sense organ in insects; therefore, most of their surface is covered with sensilla of several types. On the antennae (filiform type) of the studied *T. giganteus* specimen the sensilla—which were probably coeloconic and could be verified by an ultrastructure analysis—of only one specific type were grouped into irregularly oval fields (Figure 2A–C). Those sensillar fields appeared mainly on the third and fourth segments increasing their size from the proximal to the distal end of the segments (data not shown). The coeloconic sensilla possessed no or hardly visible pits or similar opening on their tips. Other types of sensilla, e.g., sensilla trichoidea, (Figure 2D) were less numerous. In addition to those sensilla with a visible cuticular column, the next cuticular structures were present here in a large number. Each such structure is composed of pit (or orifice) of very irregular shape (rosette-like) with grooves around each orifice (Figure 2E,F) located circularly around the base of sensilla coeloconica (Figure 2C), or also distributed outside the sensory fields (Figure 2E). These likely represent openings of the antennal gland ducts; nevertheless, they might also function as sensory organs deeply embedded into the cuticle.

*Tarsal sensilla*—*Titanus* tarsi were of the cryptopentamerous type, composed of five tarsomeres (the third was reduced and partly concealed). The first three tarsomeres were flattened, markedly massive, and broadened (Figure 3A). These three tarsomeres were covered on their underside by numerous adhesive hairs (setae) densely arranged into blocks of irregular shape (Figure 3B,C). Each adhesive hair was terminally spatulated, and covered with small spiky tubercles (Figure 3D,E). No possible secretion remainders were found on top of hairs. Hairs were firmly anchored into the tarsal base (Figure 3F).

*Other sensilla*—In insects, there are various types of sensilla and microtrichia on most parts of the body. In *T. giganteus*, a distinct row of proprioreceptive hairs (of mechanoreceptive function) was clearly visible on the anterior edge of the prothorax (Figure 4A). Further, insect mouth organs (mostly appendages) are typically covered by various sensilla (mostly of a chemoreceptoric nature). In *Titanus*, for example, sensory fields containing digitiform sensilla were present on the maxillar palps (Figure 4B). Also, labial palps bore sensory fields on their ends containing densely distributed basiconic sensilla (Figure 4C,D). Trichoid sensilla surrounded with cuticular openings were frequent on the abdominal segments (Figure 4E), and complex secretory openings were also sparsely present here (Figure 4F).

### 3.3. Internal Organs and Tissues

The age of the studied beetle was not known, however, its haemocoel was very dry with a small amount of hemolymph and fat body. The remnants of the fat body were found only on the dorsal site of the last abdominal segment. The body mass consisted mostly of musculature and the tracheal system. Trachea created a very dense network and were arranged in almost parallel bundles, each entering one spiraculum. In the abdomen each paired testis consisted of about 30 testicular follicles (Figure 5A–G) that flowed into the vas deferens. Some follicles were larger and their connection to the vas deferens was thicker (arrow in Figure 5B,C). Histological investigation verified that these follicles were ripe and produced a large number of sperm (Figure 5D–G). The vas deferens connected the testis with the vesicula seminalis where two pairs of accessory glands, the ectadenia and mesadenia, were opened. The ejaculatory duct raised from the vesicula seminalis and twisted several times before running further to the phallus. The body contained two types of musculature densely pervaded by tracheae (Figure 5H). Pink colored flight and leg muscles filled the thorax (Figure 5I; see also below) and white colored muscles were presented in the head and abdomen, responsible for movement of these body parts. In the abdomen, a thin layer of muscles fitted tightly to the dorsal and ventral cuticle of each body segment (Figure 5J,K). A yellow brown layer of pericardial cells (Figure 5J) surrounded the dorsal aorta.

In relation to body size, the brain was very small. Both hemispheres of the main brain reached a size similar to that of the paired corpora allata (Figure 6A,B). Noticeably, there were robust antennal nerves and large optic lobes, where a number of tiny axonal projections of the retinula cells run to (Figure 6A,B). The brain itself was linked via the circumesophageal connectives to the elongated subesophageal ganglion. The ventral nerve cord consisted of three larger ganglia in the thorax and four smaller ganglia in the abdomen (Figure 6C–E).

Structure of the gut corresponded to general insect gut arrangement: thicker foregut and midgut sections, and longer and thinner hindgut (Figure 6F), and clearly visible long Malpighian tubules (Figure 6G).

### 3.4. Biochemical Analyses

*Activity of digestive enzymes*—The activity of digestive enzymes was determined from the *T. giganteus* midgut and hindgut combined. The whole gut was relatively small (narrow) in comparison with the *T. giganteus* body size, probably due to the absence of food intake in *T. giganteus* adults. Despite that a certain level of digestive amylase and lipase activity was recorded in the examined gut section (Figure 7). Surprisingly, no activity of proteases was found there.

*Characterization of muscle lipids*—Curiously, the *T. giganteus* body contained practically no fat body (see above), thus, flight muscles (see Figure 5I) were used for the determination of lipidic compounds. The liquid chromatography–mass spectrometry (LC-MS) analysis of the chloroform-methanol extract taken from the muscles revealed 135 lipidic species from five lipid classes mostly present in negligible amounts (data not shown). The most abundant species from each class are shown in Figure 8A; they comprised triacylglycerols, diacylglycerols, phosphatidylethanolamines, phosphatidylglycerols, and phosphatidylcholines, of which approximately 70% was accounted for by triacylglycerols (Figure 8B). As expected, storage lipids represented approximately three quarters of muscle lipids (Figure 8C) providing corresponding energy for muscle activity.

The gas chromatography with flame ionization detection (GC-FID) analysis of the muscle lipids resulted in the identification of 13 fatty acids (FAs) possessing 12–20 carbon atoms per molecule (Figure 9A). However, just five of them were presented in amounts higher than 1%: three saturated FAs—palmitic (number of carbon atoms:number of double bonds, 16:0), margaric (17:0) and stearic (18:0), and two unsaturated FAs—oleic (18:1) and linoleic (18:2). Oleic acid was clearly the most abundant, accounting for more than 60% of all FAs in the muscles (Figure 9B).

## 4. Discussion

Due to its body length and weight *T. giganteus* indeed occupies a significant position among other giant insect species like giant scarabs *Goliathus goliatus*, *Goliathus regius*, *Megasoma elephas*, *Megasoma actaeon*, or the giant weta *Deinacrida heteracantha* [45]. There is a lot of popular-science and/or general information available concerning *Titanus* (mostly taxonomic like [6,46]*,* but a detailed scientific study of its structure and function is absent. Thus, this study brings a set of new anatomical and physiological data concerning this very interesting, and still slightly mysterious, insect species.

### 4.1. Structure of External Titanus Sensilla and Their Possible Functions

The antennae of the studied beetle were of filiform type with well-developed scape, pedicel, and flagellum. They were covered with an extremely hard cuticle, which is also typical for other parts of the *Titanus* body. The *Titanus* antennae may therefore contain a huge amount of sensilla grouped into irregularly oval fields. Those sensilla were predominantly of coeloconic type; coeloconic sensilla are widespread in insects and are known to vary in shape in various species of insects [47]. The coeloconic sensilla described here possess no, or barely visible, openings on their tips: However, we cannot exclude plugging of the tiny openings by natural secretions or secretions formed during preparation of the specimen for electron microscopy analysis. In many insects, sensilla coeloconica on the antennae serve as olfactory sensilla when a double wall and wall pores are present, or as thermo-chemosensory sensilla when wall pores are lacking [48,49,50,51]. To recognize the exact function of the described *Titanus* coeloconic sensilla, would require a detailed study of their ultrastructure. On the contrary, the trichoid sensilla, which were less numerous on the antennae, are very probably mechanoreceptors.

As mentioned above, the coeloconic sensilla are dominant on *Titanus* antennae; however, in some other cerambycid beetles, basiconical sensilla are dominant on their antennae [52].

In insects, rosette clusters of pores or single pores surrounding sensilla were described; these irregularly distributed oval or circular cuticular pores are common on insect antennae: They seem to be openings of epidermal glands [53]. We have found them also on *Titanus* antennae; the rosette-like pits of very irregular shape with grooves around orifice were circularly located around the sensilla coeloconica on the antennae (see Figure 2F), but they were also present outside the sensory fields (data not shown). They may also be openings of antennal gland ducts; nevertheless, their sensory function cannot be completely excluded, because they might represent deeply embedded cuticular sensilla. However, the latter function is less probable.

Like in other insects, in *T. giganteus* there were various types of sensilla on most parts of the body. For example, a distinct row of proprioreceptive hairs (evidently of mechanoreceptoric nature) was clearly visible on the anterior edge of prothorax (see Appendix A and Figure 4A). Most probably they control the movement of the head. A similar row of proprioreceptive hairs controls the movement of the abdominal vesicles in Archaeognatha [54]. Furthermore, and as expected, in *Titanus*, sensory fields were present on the maxillary palps (see Figure 4B). On them each trichoid sensillum was partly immersed in an elongated cuticular cavity, forming structures resembling digitiform organs found in the common cockchafer *Melolontha* [55]; the authors expected hygro-thermo-receptoric functions. Also, labial palps contained sensory fields on their end packed by densely distributed basiconic sensilla (see Figure 4C,D). Both sensilla on labial as well as on maxillar palps probably play an important role in the detection of food quality.

Pterygote insects, mainly neopteran groups, have developed specialized structures on their feet for adhering to surfaces to facilitate walking. These attachment structures can be smooth or hairy as found in many species of Coleoptera [56,57,58], Diptera [59], or Phasmatodea [60]. But a number of papers describing similar adhesive structures in various other insect groups were published within recent years. Three tarsomeres of each *Titanus* tarsus*,* flattened and broadened, were covered with numerous adhesive hairs (setae) corresponding to the adhesive hairs of the insects mentioned above. They were densely arranged into blocks of irregular shape. Such adhesive hairs differ from other types of sensilla especially because they were terminally spatulated. We did not find any rests of possible secretion on top of the adhesive hairs.

### 4.2. Titanus Central Nervous System (CNS) and Compound Eyes

Analysis of the beetle CNS structure did not bring any surprising data: The structural composition corresponded to general adult insect/beetle CNS structure, where cephalic ganglion is composed of the central brain, optic lobes laterally connected to the brain via the optic lobes, and the ventrally located subesophageal ganglion (SOG); the SOG emerges ventrally through the neck, connecting to the ventral nerve cord [61,62]. *Titanus* turned out to belong to the group of insects where the SOG is connected to the supraesophageal ganglion via a pair of stout nerve trunks named circumesophageal connectives that pass on either side of the esophagus. Further, the *Titanus* brain was characterized by large optic and antennal lobes. The size of these sensory integration centres was primarily correlated with larger complex eyes containing more photoreceptor cells and with the amount of olfactory sensory axons entering glomeruli of the antennal lobes, respectively. Recent research on another species of the Cerambycidae family, *Megacyllene caryae,* suggested that each class of odorant receptor may be correlated to a specific and unique glomerulus, and the size and placement of these glomeruli may even suggest the significance of the associated odour [63]. Whether this is connected with certain *Titanus* biology is not, in the absence of relevant data, known.

It seems that the *Titanus* compound eye is of the apposition type that is typical for diurnal insects. In the apposition eye, due to a pigment layer, the entering light reaches the retina of each ommatidium as a single spot and a mosaic-like image is composite by integrating information from the fields of view of individual facets. *Titanus* is known to fly at night, but its apposition ommatidium also suggests diurnal activity. We can neither confirm nor deny this based on available literature; nevertheless, apposition eyes have been described in several cerambycid beetles [64,65,66]. Previous study of the inner structure of the ommatidia in Cerambycidae revealed an open rhabdoms arrangement, i.e., rhabdomeres of central retinula cells are physically separated from the peripheral rhabdomeres that are segregated from each other at its proximal part [64,65,67,68]. Accordingly, a similar picture was observed in *Titanus* (see Figure 1F).

It is interesting that the facet shapes in the *Titanus* eye differed in different regions of the eye. Facets are regular in the central area and irregular on the periphery of the eye. A similar arrangement has been observed in staphylinid beetles *Creophilus erythrocephalus* and *Sartallus signatus* [69], in the erotylid fungus beetle *Neotriplax lewisi* [70], and in the small diving beetle *Agabus japonicus* [71]. Generally, the presence of many irregularly shaped facets suggests decreased resolution, that most likely allows the beetle only to distinguish light intensity and/or to refer to the position of the sun, which is important for orientation [70].

Originally, during arthropod evolution, each segment of the body contained one ganglion of the ventral nerve cord [61,72]. However, the general evolutionary trend has led to the gradual fusion of neighboring ganglia and significant reduction of their total number [72]. Phylogenetic studies of diversity in ganglion fusion have revealed that coleopterans are very diverse, varying from the fusion of all thoracic and abdominal ganglia into one to three distinct thoracic ganglia and a maximum of eight abdominal ganglia, respectively [73,74,75]. In Cerambycidae, fusion of the last thoracic ganglion with the first two abdominal ganglia has been described, resulting in three thoracic and four abdominal ganglia [76]. Accordingly, we have found the same picture in the ventral nerve cord of *Titanus*.

### 4.3. Titanus Gut and Fat Body

It is supposed that *T. giganteus* adults, as is common for Prioninae representatives, do not feed—thus the gut of the studied beetle was largely atrophied. Such atrophy and degeneration of the gastrointestinal tract during metamorphosis in the pupal and adult stages was also described by Benham [32] in another representative of the Prioninae subfamily, *Prionus laticollis*. Interestingly, in the anterior midgut of one of the 33 tested *P. laticollis* adults a wood-like material was observed, probably as a residue from the larval period. However, unlike *Titanus*, the *P. laticollis* gut was surrounded with an extensive fat body. It seems to be logical that atrophy of the gastrointestinal tract increases with time elapsed since eclosion [32]. This might explain the almost complete absence of a fat body in the studied *Titanus,* and its presence in *P. laticollis*. Our beetle was tested on the tenth day after capture, which is close to the end of its life (estimated as 14 days), and therefore the fat body was already metabolically exhausted.

Another hypothetical explanation for the almost complete absence of a *Titanus* fat body could be the location of energy sources closer to the target point of consumption—its massive muscles—in the large insect body, whose size approaches physiological limits. Hypothetically, it might be possible that a giant insect organism no longer relies on “excessive” metabolic processes in adulthood and that adult fat bodies are not properly formed. It would be interesting to perform a test to this effect on an individual immediately after eclosion: the general rule for holometabolic insects is that during the pupal stage the fat body disintegrates and adult fat bodies are regenerated at the end of the pupal period [77].

Surprisingly, *Titanus* larvae have never been scientifically described, thus, nothing is known about its feeding strategy. Nevertheless, it is supposed the *Titanus* larva feeds on dead wood, and so, the enzymatic activity recorded in our adult beetle probably represented residual larval digestive processes. Generally, xylophagy is a complicated matter that predominantly includes activity from carbohydrases, amylases, cellulases, hemicellulases, polygalacturonases, and other similar enzymes typical for wood digestion. Such enzymes have been described in detail, e.g., in the cerambycid *Pcacothea hilaris* [16], which feeds on mulberry wood and leaves. Also, generally it was shown a long time ago [78] that Cerambycidae representatives are able to digest starch, hemicellulose, and cellulose; it is supposed that especially the digestion of the two latter items depends mostly on symbiotic organisms. Interestingly, we recorded relatively higher amylase activity in the *Titanus* gut. Nevertheless, no protease activity was recorded there, however, proteases have been repeatedly reported in adult cerambycids: *Enaphalodes rufulus* [79], *Osphranteria coerulescens* [80], and *Fulvum villers* [81]. Phylogenetic aspects of the activity of cysteine and serine proteases in eight cerambycid beetles were studied in detail by Johnson and Rabosky [18]. Several explanations are available for the absence of proteinase activity in the adult *Titanus* gut. Protease activity is highly dependent on gut pH [80]; alkaline environments are assumed to be optimal for the gut of Cerambycidae representatives [17,18,19]. Therefore, a slightly alkaline pH was used in our protease experiments, nevertheless, due to a limited amount of the tested material, a detailed study of optimal pH determination could not be performed. Therefore, some protease activity under higher pH conditions cannot be completely excluded although it is not very likely. Another explanation for the absence of protease might be the full utilization of proteins predicted in the second half of the adult phase of the *Titanus*’s life, when there is no need for any further development or growth, and thus no need for protease activity.

Obviously, it would be interesting to repeat the tests on a larger number of individuals to describe the composition of digestive enzymes optimally at different stages of life, and also to examine protease activity in females which may have other needs for anabolic processes. Nevertheless, due to the problematic availability of *Titanus* beetles this is not realistic.

### 4.4. Titanus Muscles and Their Lipid Content

Insect flight muscles are a very metabolically active organ capable of increasing their metabolism between 50- to 100-fold during flight [82]. Such an enormous performance requires an intensive supply of energy substrates, either carbohydrates or especially for long-distance or intensive flights, lipids. In insects, lipids are stored mainly in the fat body as triacylglycerols (TGs), and are transported as diacylglycerols (DGs) via hemolymph into the muscles, where they are utilized in response to bioenergetic demands. Some TGs are also stored in other organs including the muscles. Accordingly, also in *Titanus* TGs were present at a relatively high level in the flight muscles, whereas the DG level as well as the level of structural lipids (that are important components of the cell membrane) were much lower. It is known that the majority of insect fatty acids (FAs) is represented by just 8–9 FAs possessing 12–18 carbons per molecule. Indeed, in *Titanus* the most abundant FAs were those with 18 and 16 carbons; their abundance in the flight muscles decreased in the order: oleic acid (18:1) > palmitic acid (16:0) > stearic acid (18:0) > linoleic acid (18:2). Interestingly, a similar FA composition with the same quantitative order was also recorded in the tenebrionid beetle *Zophobas atratus* [83]. The highest level of oleic acid was recorded not only in the beetles, but also in the locust *Locusta migratoria* [84] and the blood-sucking bug *Panstrongylus megistus* [85]. *Titanus* muscles also contained the uncommon margaric FA (17:0), and several polyunsaturated FAs. The latter FAs are usually necessary in small proportions in the body, and in *Titanus* they included arachidonic acid (20:4, n-6), and a mixture of two C18:3 FAs—α-linolenic (C18:3, n-3) and γ-linolenic (C18:3, n-6). The importance of arachidonic acid in insect metabolism is discussed in a review paper [86]. This acid is an important component of membrane phospholipids and thus affects the structure of cellular and subcellular membranes. Arachidonic acid is also a precursor of prostaglandins and other eicosanoids and thereby affects various aspects of insect regulatory physiology. We can speculate that these FAs are probably derived from diet, where they might be products of wood-decaying fungi.

### 4.5. Titanus Testes

The male reproductive organs of *Titanus* were represented by a pair of multifollicular testes situated laterally in the proximal part of the abdomen. A general description of the reproductive system of *Titanus* corresponded to the male system of another Prioninae, *Prionoplus reticularis* [30]. The pupal testis in *P. reticularis* consisted of 12–15 lobes, each containing 15 follicles. Edwards [30] compared the reproductive organs of Prioninae and Lamiinae with the conclusion that in adult non-feeding species, gametogenesis is terminated at the pupal stage and gonadal degeneration occurs during adult life. The adult *Titanus* displayed a significantly smaller number of follicles, which can be attributed to interspecific variability or degeneration of the gonads in the aged adult.

It was interesting that some of the follicles were larger than others in our *Titanus* specimen. Histochemical data confirmed increased spermatogenesis within these follicles. Spermatogenesis is characterized by the formation of the sperm bundle. After spermatogenesis within the follicle, separate spermatozoa are released into the diverticula of the vas deferens. Here, they become attached to central rods (spermatostyles), forming secondary conjugates (spermiozeugmata). A similar testicular arrangement has been found in other species e.g., in male scarabaeid beetles *Phytalus sanctipauli* [87].

## 5. Conclusions

This paper presents the general results of our study focusing on the internal/external morphology and physiology of *T. giganteus*. Current knowledge about *Titanus* biology was enriched with interesting information about the structure/ultrastructure of sensilla on the antennae, legs, and abdomen, the arrangement of compound eyes, and the structure of various internal organs including the central nervous system. Basic data about lipid energy metabolites in the muscle system were obtained as well. It can be hypothesized from the morphology, and analyses of the digestive and reproductive tract (intestinal atrophy, the almost complete absence of fat bodies, probably minimal protease activity, and probably degeneration of gonads) that adult *Titanus*—despite its size and logically greater metabolic needs—relies on previously accumulated reserves rather than feeding from the time of eclosion. This set of findings represents a solid base for future investigation of *Titanus* morphology and physiology that could help to elucidate the upper limits of physiological processes in insects.

## Figures and Tables

**Figure 1 insects-11-00120-f001:**
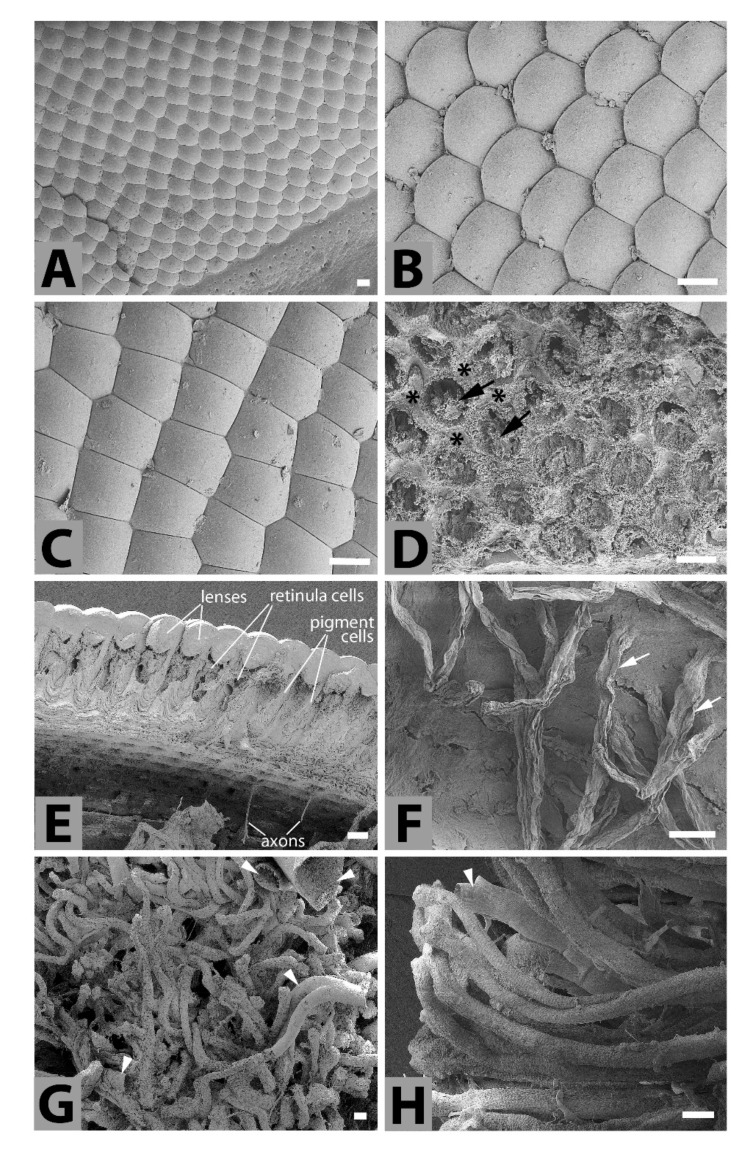
Scanning electron microphotographs of *T. giganteus* compound eye. (**A**)—Part of the compound eye. (**B**,**C**)—The regular and irregular shapes of facets, respectively. (**D**)—Dorsal view to the ommatidia after removing of the lenses. Black arrows depict the retinula cells and black stars the pigment cells. (**E**)—The cross section through the compound eye containing ommatidia with biconcave lenses, retinula cells, pigment cells and the bare axons extending from the rhabdoms. (**F**)—The inner surface of the compound eyes covered by basement membrane with extending optical fibers (white arrows) where the axonal bundles are surrounded by the sleeve of supporting cells. (**G**)—Tangle of muscles and tracheas (arrowheads) intertwine between the optical bundles and fulfil the inner hollow of the compound eyes. (**H**)—The detail image of a few muscle bundle and tracheas (white arrowhead). Scale bar: 100 μm.

**Figure 2 insects-11-00120-f002:**
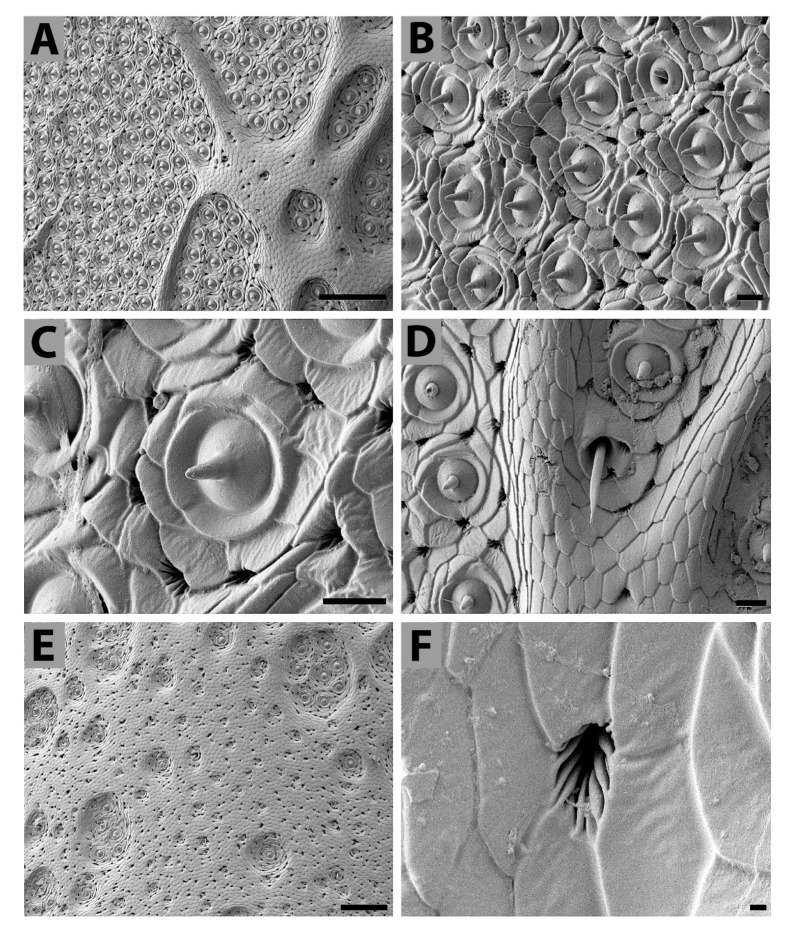
Scanning electron microphotographs of *T. giganteus* antenna. (**A**)—General view on the coeloconic sensillar fields. (**B**)—Detail of one sensillar field. (**C**)—Individual coeloconic sensillum (surrounded with several rosette-like openings). (**D**)—Two different types of antennal sensilla: trichoid sensillum (middle of the picture) and coeloconic sensillum. (**E**)—Antennal cuticle outside of the sensillar field with numerous openings and individual or grouped coeloconic sensilla. (**F**) Detail view of the rosette-like opening. Scale bar: 100 μm (**A**,**E**), 10 μm (**B**–**D**); 1 μm (**F**).

**Figure 3 insects-11-00120-f003:**
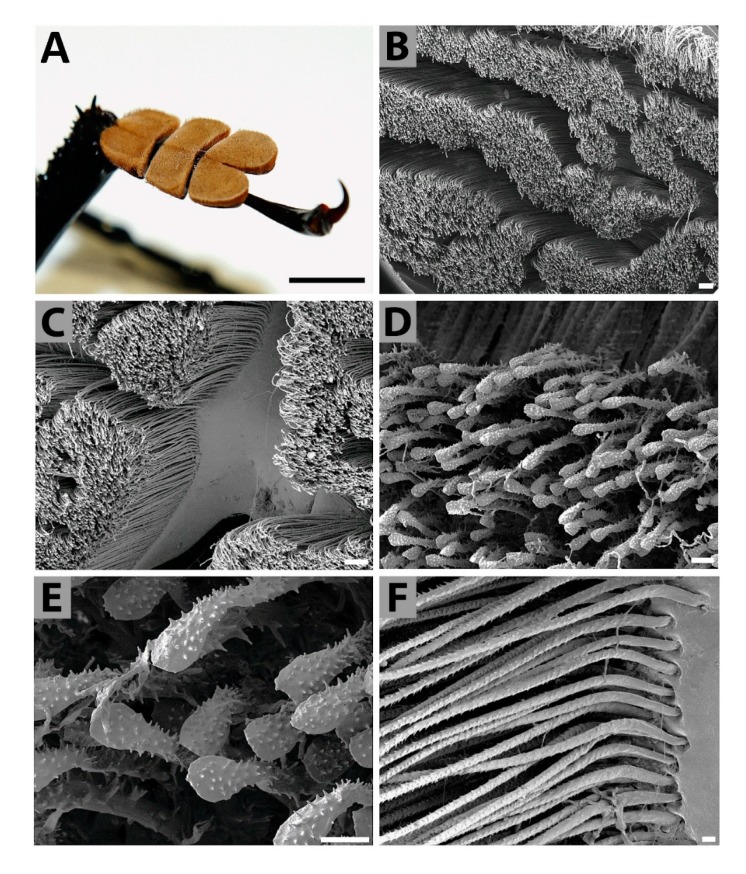
Macrophoto (**A**) and scanning electron microphotographs (**B**–**F**) of *T. giganteus* tarsus. (**B**)—General view on the tarsal hairs grouped in wide and irregular rows. (**C**)—Detail of the tarsal hairs field. (**D**)—Semi-detail of individual adhesive tarsal hairs. (**E**)—Full detail of the adhesive tarsal spatulate hairs. (**F**)—Attachment of the adhesive tarsal hairs to the tarsal base (cuticle). Scale bar: 5 mm (**A**); 100 μm (**B**,**C**); 10 μm (**D**–**F**).

**Figure 4 insects-11-00120-f004:**
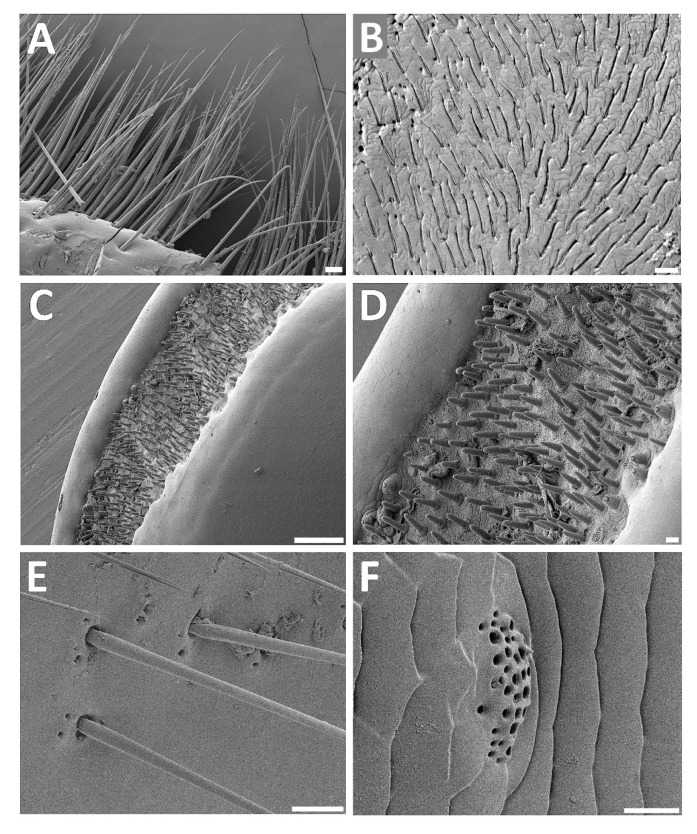
Scanning electron microphotographs of selected parts of *T. giganteus* body. (**A**)—Mechanoreceptor (proprioreceptive) hairs on the anterior edge of prothorax. (**B**)—Sensory field on the maxillar palp (digitiform sensilla with sensory parts immersed in elongated cuticular cavities). (**C**)—Sensory end of the labial palp. (**D**)—Sensory end of the labial palp—detail. (**E**)—Abdominal trichoid sensilla surrounded with openings. (**F**)—Detail of the abdominal complex secretory opening. Scale bar: 100 μm (**A**,**C**,**E**); 30 μm (**B**); 10 μm (**D**,**F**).

**Figure 5 insects-11-00120-f005:**
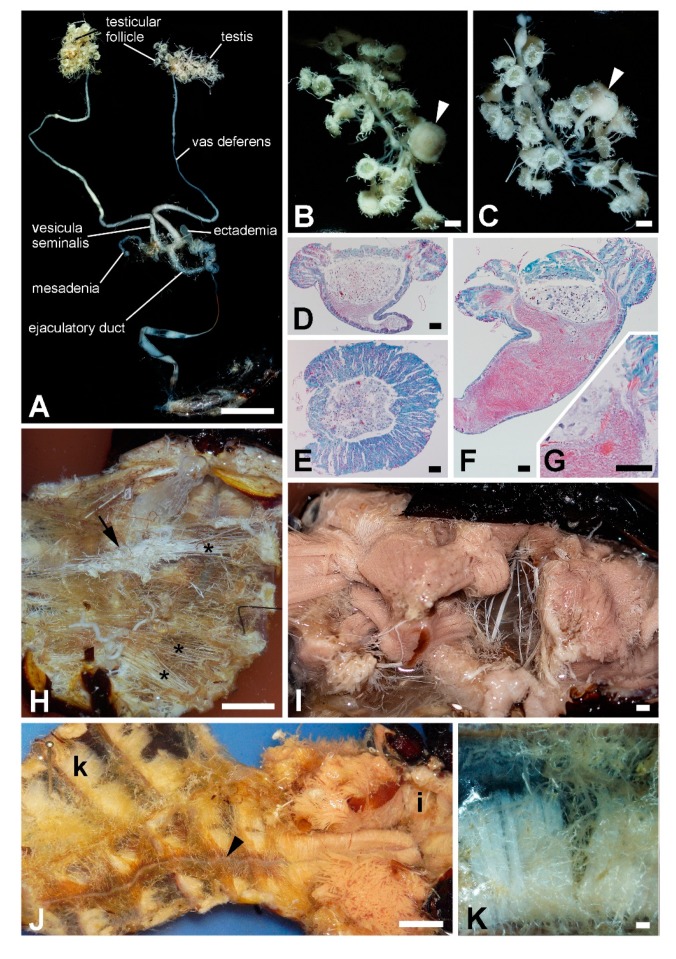
*T. giganteus*, internal morphology. (**A**)—Male reproductive system. (**B**,**C**)—Detailed images of testis. Arrowheads depict one testicular follicle that is larger than all others are. (**D**,**E**)—Mallory staining of longitudinal- and cross-section, respectively, through small testicular follicle. (**F**)—Mallory staining of longitudinal-section of large testicular follicle with large number of sperm (in red). (**G**)—The detailed view on the sperm clusters (red). (**H**)—The open abdomen with bundles of trachea (stars) and testes (arrow). (**I**)—Flight muscles in the thorax. (**J**)—Muscles in the abdomen and thorax. Lower-case letter show areas of detailed images (I) and (**K**). The dorsal aorta is marked by arrowhead. (**K**)—Detailed image of muscle bundle under the cuticle on ventral thorax. Scale bar: 1 cm (**A**,**H**,**J**); 1 mm (**B**,**C**,**I**,**K**); 100 µm (**D**–**G**).

**Figure 6 insects-11-00120-f006:**
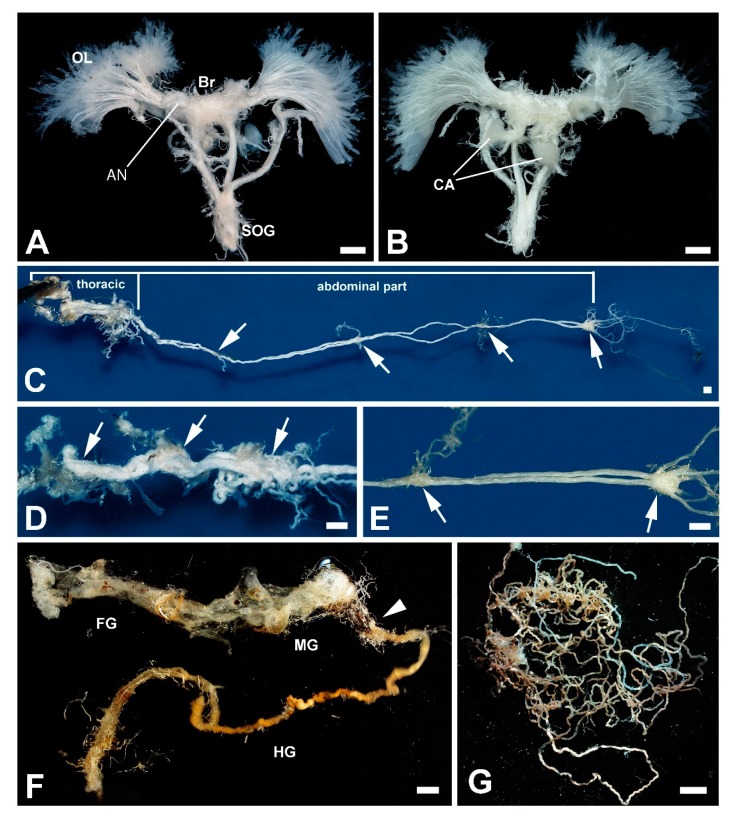
*T. giganteus*, central nervous and digestive systems. (**A**,**B**)—Anterior and posterior view of the cephalic ganglion, respectively. AN, antennal nerve; Br, brain; CA, corpora allata, OL, optic lobe, SOG, subesophageal ganglion. (**C**)—Ventral nerve cord divided to the thoracic and abdominal part. The arrows show ganglia in the abdominal part. (**D**)—The detail image of thoracic part of the ventral nerve cord. The arrows show ganglia. (**E**)—The detail image of posterior region of the abdominal part of the ventral nerve cord. The arrows show ganglia. (**F**)—The empty gut. FG, foregut; MG midgut; HG, hindgut. The white arrowhead shows the place where the Malpighian tubules are attached. (**G**)—Malpighian tubules. Scale bar: 1 mm.

**Figure 7 insects-11-00120-f007:**
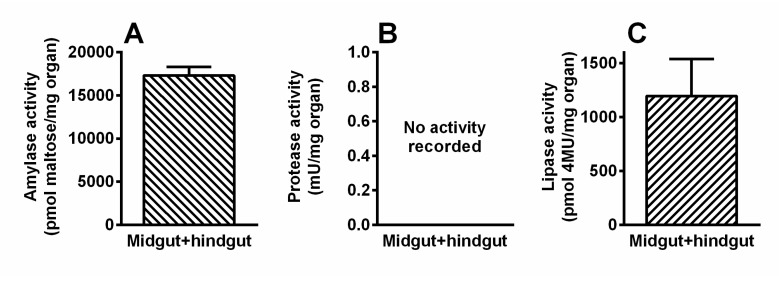
Levels of amylase (**A**), protease (**B**) and lipase (**C**) activities in *T. giganteus* gut. The bars represent mean ± SD of five (technical) replicates (n = 5).

**Figure 8 insects-11-00120-f008:**
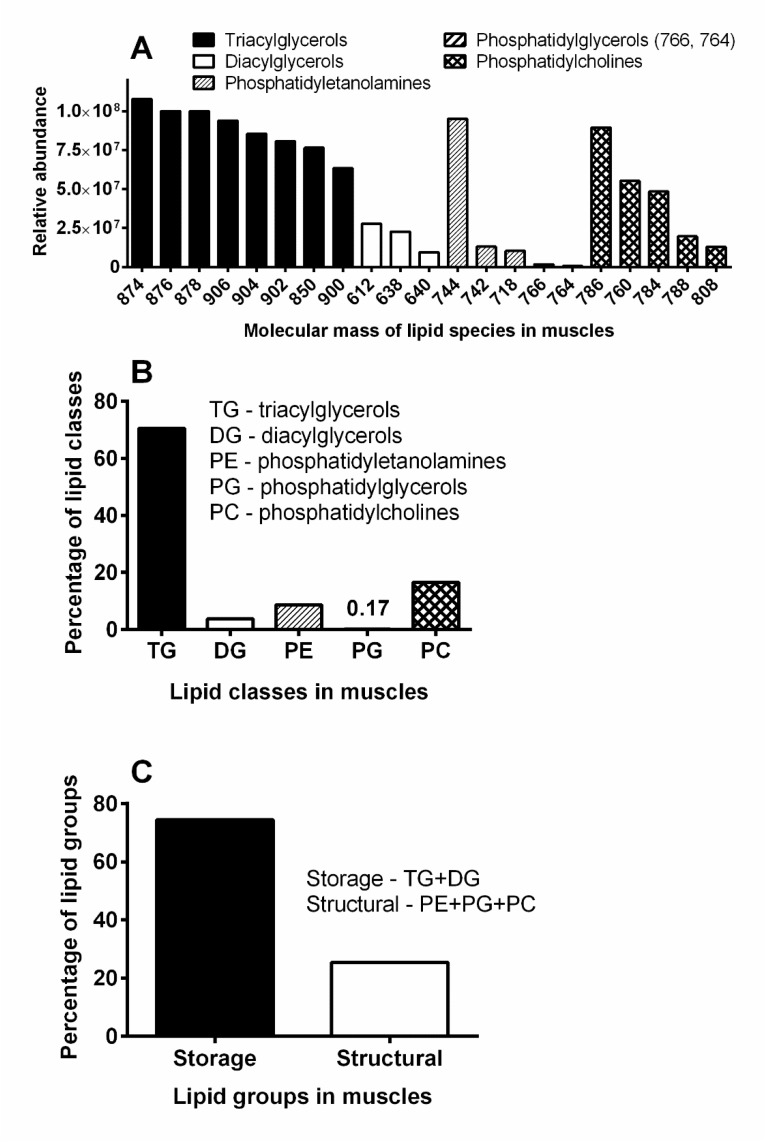
LC/MS analysis of lipids from *T. giganteus* muscles. (**A**)—Major lipidic species. (**B**)—Major lipidic classes. (**C**)—Ratio between storage and structural lipid groups.

**Figure 9 insects-11-00120-f009:**
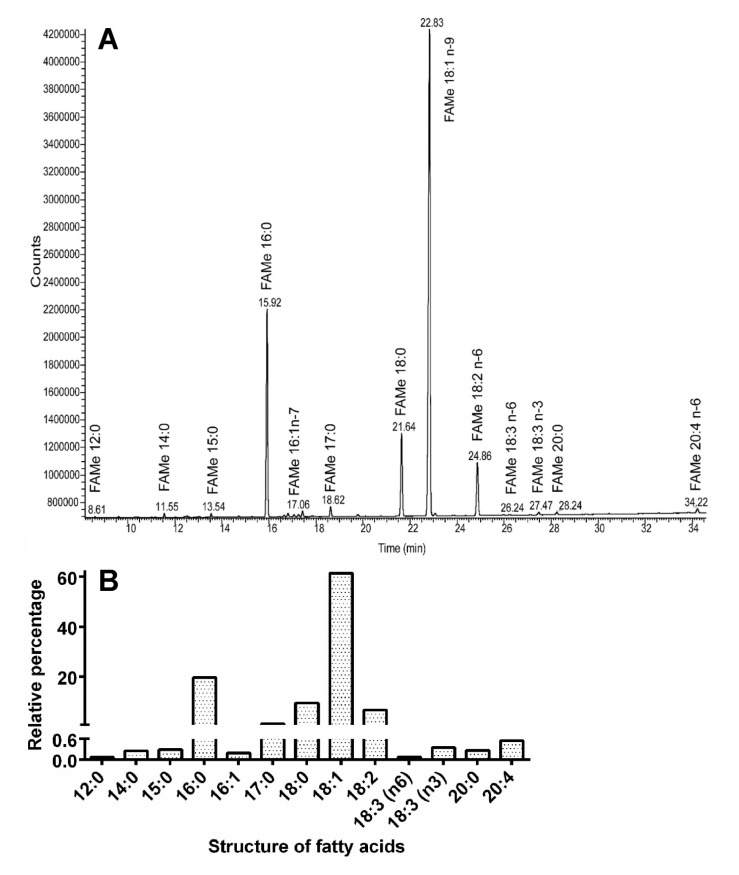
Analyses of fatty acids from *T. giganteus* muscle lipids. (**A**)—Chromatogram recorded by gas chromatography with flame ionization detection (GC-FID) shows the most abundant fatty acids (FAs). (**B**)—Relative percentage of those FAs.

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
