# Peer review of "First Comprehensive Study of a Giant among the Insects, Titanus giganteus: Basic Facts from Its Biochemistry, Physiology, and Anatomy"

_insects, 2020, doi:10.3390/insects11020120_

Round 1
Reviewer 1 Report
Authors at first present a comprehensive morphological / anatomical as well as physiological / biochemical study on one of the largest insect worldwide. Although only one male specimen (close to the end of adult life) of the "gigant" was available for the present study, which naturally restricted the possible experiments, the results found are absolutely worth publishing. The microphotographs and electron microscopic pictures are of high quality and the manuscript is well written. The most surprising result for me was the fact that such a large beetle seems to exclusively live from larval energy stores in its adult stage (at least the male) because its gut was completely empty and no fat body could be found. Another surprising result is the presence of C 17:0 fatty acid in the flight muscle fat. All these unexpected observations are in depth discussed by the authors. Another unexpected result is the increased spermatogenesis (plenty of sperm) in the old adult male, which could be discussed in more detail.
Minor corrections:
line 109: staining was performed line 141: what does the "ss" mean? line 149: give species name in italics line 171: there is no chapter 2.6.1 in the manuscript lines 218/219: this statement should be part of the Discussion Fig. 1: the black stars in D are hardly to be seen line 273: please check whether ectademia and mesodenia is correct line 290: scale bar for D, E, F 100 μm? line 741: give species name in italics line 757/758: Annu. Rev. Entomol. is correctAuthor Response
We would like to thank both the anonymous reviewers for their corrections and suggestions – all changes suggested were incorporated into the MS text or clarified below. Additionally, the text was checked by the Spelling check function and minor corrections were done. All changes are highlighted using the Track changes function.
Reviewer 1
Dear Authors,
congratulations for preparation very interesting manuscript.
I have only a few suggestion.
In abstract: the "brain was small" and "the large number of sperm" sounds a little bit dull in scientific article, so try to use other words.
Reply: the wordings were improved
M&M
Please, add information about the conditions of samples separation by HPLC as concentration of eluents, isocratic/gradient/, the flow ratio, temperature etc.
Reply: the information was added into the text
line 149. the "T. giganteus" should be in italic.
Reply: corrected
Results:
line 266: change "fat" into "fat body"
Reply: corrected
Have you taken a photo of sperm cells? Or is it still possible to do it? It will be great to add this information, also the size of sperm cells.
Reply: corresponding photo was added – see Figure 5G
Can you add some "map" of distribution of ommatidia types in the eye?
Reply: Unfortunately we do not have these data
Kind regards,
reviewer
Reviewer 2 Report
Dear Authors,
congratulations for preparation very interesting manuscript.
I have only a few suggestion.
In abstract: the "brain was small" and "the large number of sperm" sounds a little bit dull in scientific article, so try to use other words.
M&M
Please, add information about the conditions of samples separation by HPLC as concentration of eluents, isocratic/gradient/, the flow ratio, temperature etc.
line 149. the "T. giganteus" should be in italic.
Results:
line 266: change "fat" into "fat body"
Have you taken a photo of sperm cells? Or is it still possible to do it? It will be great to add this information, also the size of sperm cells.
Can you add some "map" of distribution of ommatidia types in the eye?
Kind regards,
reviewer
Author Response
We would like to thank both the anonymous reviewers for their corrections and suggestions – all changes suggested were incorporated into the MS text or clarified below. Additionally, the text was checked by the Spelling check function and minor corrections were done. All changes are highlighted using the Track changes function.
Reviewer 2
Authors at first present a comprehensive morphological / anatomical as well as physiological / biochemical study on one of the largest insect worldwide. Although only one male specimen (close to the end of adult life) of the "gigant" was available for the present study, which naturally restricted the possible experiments, the results found are absolutely worth publishing. The microphotographs and electron microscopic pictures are of high quality and the manuscript is well written. The most surprising result for me was the fact that such a large beetle seems to exclusively live from larval energy stores in its adult stage (at least the male) because its gut was completely empty and no fat body could be found. Another surprising result is the presence of C 17:0 fatty acid in the flight muscle fat. All these unexpected observations are in depth discussed by the authors. Another unexpected result is the increased spermatogenesis (plenty of sperm) in the old adult male, which could be discussed in more detail.
Minor corrections:
line 109: staining was performed
Reply: corrected
line 141: what does the "ss" mean?
Reply: corrected
line 149: give species name in italics
Reply: corrected
line 171: there is no chapter 2.6.1 in the manuscript
Reply: corrected
lines 218/219: this statement should be part of the Discussion
Reply: the statement was deleted. Similar one is in Discussion
Fig. 1: the black stars in D are hardly to be seen
Reply: the size of the stars was increased
line 273: please check whether ectademia and mesodenia is correct
Reply: corrected
line 290: scale bar for D, E, F 100 μm?
Reply: corrected
line 741: give species name in italics line
Reply: corrected
757/758: Annu. Rev. Entomol. is correct
Reply: corrected